# Implementing a Synchronization Method between a Relational and a Non-Relational Database

**Cornelia A. Győrödi** *[ID], **Tudor Turtureanu, Robert Ş. Győrödi** *[ID] **and Doina R. Zmaranda** [ID]

Department of Computers and Information Technology, University of Oradea, 410087 Oradea, Romania;
ttudor93@gmail.com (T.T.); dzmaranda@uoradea.ro (D.R.Z.)
* Correspondence: cgyorodi@uoradea.ro (C.A.G.); rgyorodi@uoradea.ro (R.Ş.G.)

**Abstract:** The accelerating pace of application development requires more frequent database switching, as technological advancements demand agile adaptation. The increase in the volume of data and at the same time, the number of transactions has determined that some applications migrate from one database to another, especially from a relational database to a non-relational (NoSQL) alternative. In this transition phase, the coexistence of both databases becomes necessary. In addition, certain users choose to keep both databases permanently updated to exploit the individual strengths of each database in order to streamline operations. Existing solutions mainly focus on replication, failing to adequately address the management of synchronization between a relational and a non-relational (NoSQL) database. This paper proposes a practical IT approach to this problem and tests the feasibility of the proposed solution by developing an application that maintains the synchronization between a MySQL database as a relational database and MongoDB as a non-relational database. The performance and capabilities of the solution are analyzed to ensure data consistency and correctness. In addition, problems that arose during the development of the application are highlighted and solutions are proposed to solve them.

**Keywords:** NoSQL; MySQL; MongoDB; database synchronization; database replication

## 1. Introduction

Databases play an essential role in managing and storing data in an application. Two of the most popular and widely used types of databases are relational and non-relational.

Relational databases are effective in handling structured data and complex relationships between tables. But, because relational databases have a more restrictive and rigid data model and, sometimes, some limitations when scaling, a different approach to data storage, namely NoSQL (Not Only SQL (Structured Query Language)) databases, has emerged [1–3]. Unlike SQL databases, NoSQL databases are not based on tables, using documents or other models, therefore being more flexible and having capabilities of handling large volumes of unstructured or semi-structured data [4,5].

Therefore, to cope with the large volume of data, many companies that formerly used only relational databases in their operations have decided to migrate to NoSQL databases to solve certain operations that require time [6].

By synchronizing these two types of databases, one can benefit from the advantages of both [7]. As stated in [8], in massive scale and high concurrency applications like search engines relational databases are complemented by specifically designed NoSQL databases. Although these two types of databases have different approaches to data storage and access, there are situations in practice where migration or replication is necessary to exploit the specific capabilities of each database [6].

By synchronizing a relational database with a non-relational database, one can use the scalability and performance capabilities of the non-relational database for specific operations, such as searching or displaying data, while the relational database can be

used for more complex query operations or to ensure data integrity. For example, in a typical sales application, a relational database could represent the best solution for storing data about products, users, orders, etc. But, if the application has, for example, a recommendation system that wants to suggest similar products available after a search, a non-relational database that has the ability to connect nodes together for faster navigation could allow fast search of chained products; moreover, it could provide short search time between objects without links when used in searches for products by different keywords. In such an approach, synchronization between the two databases becomes an important issue.

Another situation is the transition from one model to another. Sometimes an organization may have an existing relational database and wants to move to a non-relational model to meet scalability or flexibility needs. By synchronizing these two databases, data and applications can be gradually transferred to the new non-relational database without affecting functionality and existing users. This process has to allow for a smoother transition and reduces the risks associated with completely changing the data infrastructure.

In this context, the paper proposes a practical IT approach to this problem and tests the feasibility of the proposed solution by developing an application that maintains the synchronization between the MySQL database as a relational database and MongoDB as a non-relational database.

The paper is organized as follows: The Section 1 contains a short introduction emphasizing the motivation of the paper, followed by Section 2 which reviews related work. The description of the implementation, the solution architecture, and the system configuration used in this work are described in Section 3. The experimental results and their analysis regarding the latency, data consistency, and operations replication performance on the two databases in an application that uses large amounts of data are presented in Section 4. Finally, some conclusions regarding the analysis are revealed.

## 2. Related Work

Many studies have been carried out in recent years on replication solutions of a relational database with a non-relational database, but search efforts have produced no works specifically addressing the synchronization of a relational database with a non-relational database.

In [4], the authors address MySQL and Elasticsearch's performance as databases in the case where an application's data are replicated. In [9], a way to integrate two relational databases for the same application is presented. The method implies translating XQuery queries into a 'mediator' program which then applies these queries to the databases.

Some papers address the ways to preserve relationships when inserting data into a non-relational database. In [6], a solution is provided to process the relational database schema, which is to insert it into a columnar non-relational database. In [10], the authors define solutions to map relations from a relational database to a non-relational one. They go through all types of relationships, one-to-one, one-to-many and many-to-many, and explain the construction of the format of the data using JSON to achieve an equivalent structure in the non-relational database. In [11], an alternative mapping solution is proposed, suggesting the utilization of a list of IDs within the NoSQL database field to establish associations between entities in cases of many-to-many relationships.

In [12], the authors describe the synchronization of two databases in a heterogeneous environment, however, the authors talk about an environment that contains two relational databases: Oracle and MSSQL. The method implies that changes in one database are captured, processed, and then applied in the other database. After that, a correlation between the data types and functions is made for the two databases. Here, capturing statements to a database is achieved by intercepting the SQL code.

The work presented in [13] details a functional system that replicates a MySQL database to a MongoDB database. Although it is a comprehensive solution that also addresses more advanced mechanisms than CRUD operations, such as triggers, indexes,

and integrity constraints, data migration solely facilitates the unidirectional replication of data. Synchronization requires changes to flow in real time and in both directions.

Various papers related to non-relational databases were found. Reference [14] describes a pipeline to collect, prepare, and store Big Data. In [15], the authors show how Big Data analysis helps companies make more informed decisions, based on the data that are produced and what tools can be used for the purpose of analysis. In [16], the advantages of using a non-relational database when developing an application are presented. It also highlighted why non-relational databases will be used more and more in the future.

In this idea, the method of synchronization between MySQL as a relational database and MongoDB as a non-relational database proposed in this paper becomes a promising research direction, due to the increasing amount of data to be processed and queried as fast as possible.

## 3. Description of the Implementation

The testing architecture's centerpiece is a Java-based application that executes synchronization logic. MySQL [17] was used as a relational database and MongoDB [18] was used as a non-relational database. MongoDB is the most popular type of NoSQL database, with a continuous and steady rise in popularity since its launch [7]. It is a cross-platform, open-source NoSQL database that is document-based (which is written in C++), completely schema-free, and manages JSON-style documents [16].

During the actual implementation, Debezium [19] was used, a CDC (Change Data Capture) platform that looks in database logs, to detect changes, and then writes the details of the changes to a JSON that is then put on a Kafka [20] topic. Synchronization logic, implemented in Java [21], implies reading the Kafka messages and applying the changes to the databases. For portability, Kafka [22], Debezium [23], and their dependencies are installed as Docker [24] images.

### 3.1. The Solution Architecture

The proposed solution has the following components:

- The Java application, which has the role of receiving and processing messages through Kafka queues. The processing consists of reading the message, determining the operation type from the op field, then applying an insert, update, or delete using the data from the data field. The application's connections are with Apache Kafka [22], MongoDB [18] and MySQL [17].
- Apache Kafka, whose role is primarily to provide an environment for transmitting messages in a robust way, in which information is not lost and a large volume of data can be processed. Through Debezium [23] connectors, changes in databases are detected and messages are posted on Kafka topics.
- The relational database, MySQL [17] to which the Java [21] application connects, the Debezium connector for MySQL and, in a business environment, a client application that makes changes to the data.
- The non-relational database, MongoDB, to which the Java application connects the Debezium connector for MongoDB and, in a business environment, a client application that makes changes to the data.

These components and connections are highlighted in Figure 1.

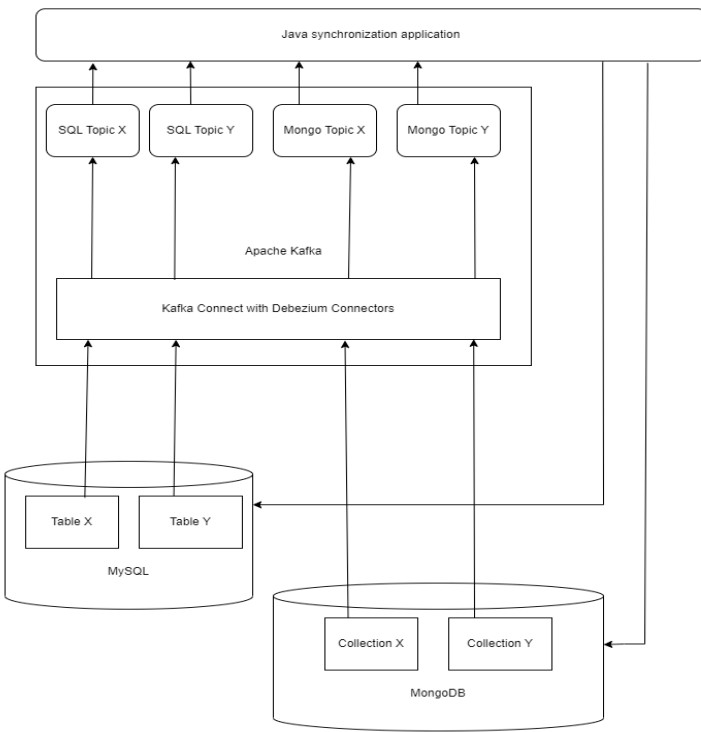

**Figure 1.** Application architecture.

*3.2. Application Description*

The Java synchronization application is a Spring Boot-based application that uses Spring Data and the Spring Boot Starter Web package, along with Kafka Streams and Jackson Core. When the application is started, the Kafka topics are retrieved from Kafka, one topic corresponds to one table or collection. A processor for each topic is instantiated, and the processor then listens for messages on its topic. When a message is consumed, the payload is deserialized, the operation is detected as Debezium writes it on the message, and depending on that, that operation is then reapplied to the other database through the persistence layer.

If the operation fails, a retry mechanism will re-attempt it a configurable number of times using exponential backoff as a backoff strategy. A common reason for failure is when trying to synchronize two or more inserts that have a foreign key constraint between each other. If the dependent row is inserted before the row it is depending on, the application will throw a *ConstraintViolationException*. Notable checks are made both in the processor and in the persistence layer for redundant operations. For example, it will not insert an already inserted row, delete an already deleted row or update a row using its current data. These checks are carried out to detect and avoid infinite loops. In case an application erroneously inserts into MongoDB an entry that has a primary key that already exists in that table, the operation will fail due to a *ConstraintViolationException*, and the retry mechanism will eventually stop. The same case does not apply to MySQL, since the operation will fail before the insert is executed.

3.2.1. Primary Key Synchronization Problem

The problem involves the databases that are kept in synchronization, which face a conflict when they simultaneously insert a record with the same ID but different data. This creates a collision and leads to errors, as the system cannot distinguish between the two entries. The system needs to ensure that the same ID is not used to represent different records in the two databases.

The first solution involves using Globally Unique Identifiers (GUIDs) for all IDs in the databases. GUIDs are 128-bit numbers that are designed to be unique across all systems,

thus practically eliminating the risk of duplicate IDs. Since the two databases would now generate their own unique IDs, the chance of a collision between them would be minimized.

The second solution, assigning an ID range to each database involves allocating specific ranges of IDs to each database, ensuring that they will not overlap. For instance, the first database could use IDs from 1 to 1000, while the second uses 1001 to 2000.

The third solution involves configuring each database to increment its IDs by the same amount but starting them at different values, such as even numbers for one database and odd numbers for the other.

Of the three solutions, assigning GUIDs to all IDs is the most attractive option. While it may entail more storage and some complexity in implementation, it offers a scalable and robust solution. In contrast, the ID range solution would require database clients to coordinate with the application for ID ranges, adding unnecessary complexity. The incrementing solution, although elegant, requires extra configuration. The first two solutions only work with integer IDs. Using GUIDs provides the most reliable way to ensure that both databases can insert unique records without collisions, making it the best solution.

### 3.2.2. Update Collision Resolution

Another situation where two operations would cause ambiguous results is when an update in one of the databases is made at the same time as an update in the other database on the same row. One strategy could be 'last one wins', where the update that has the latest timestamp is persisted. Another strategy is to 'merge effects', which only applies to updating amounts. For example, in an account balance, one update adds USD 10 and another adds USD 20. After the collision is detected and resolved, the total amount should be USD 30 higher than before the updates. We chose the first strategy for our application.

### 3.3. System Configuration

For the whole ensemble to work, some notable configurations need to be made. First, MongoDB has to run as a replica set, or it will not write its changes to the *oplog*. This is important as Debezium looks for changes in Mongo's *oplog*. For Debezium to have access to MySQL, it will need a user with all read rights. When creating the Debezium workers using Kafka Connect, it will need the name of the database, and the prefix it will use to name topics. For the MongoDB connector, the *capture.mode* setting needs to be set to *change_streams_update_full_with_pre_image* or else the messages for updates will not contain both the data before and after the update. This is important when avoiding infinite loops.

The structure of the relational database used in this paper is composed of three entities: *flight*, *ticket*, and *passenger*, as shown in Figure 2.

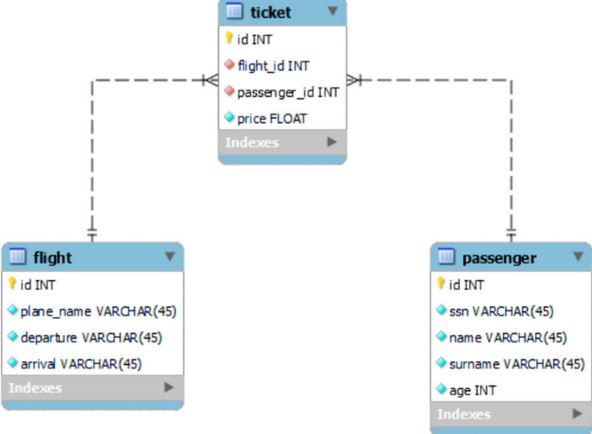

**Figure 2.** MySQL database structure.

For MongoDB, the structure is described in Figure 3.

```
Flight
{
  "_id": 3,
  "plane_name": "Airbus A380",
  "departure": "Rome",
  "arrival": "Paris"
}
Passenger
{
  "_id": 6,
  "ssn": "1911021313724",
  "name": "John",
  "surname": "Doe",
  "age": 47
}
Ticket
{
  "_id": 2,
  "flight_id": 3,
  "passenger_id": 6,
  "price": 309.11
}
```

**Figure 3.** MongoDB database structure.

## 4. Method and Testing Architecture

To test the application, two sets of REST APIs were used: one for MongoDB and one for MySQL. Several aspects of the application were tested: synchronization, latency, performance in batch operations. How long it takes to perform the synchronization was measured according to the number of processed entities. The ability to synchronize tables bound by FOREIGN KEY constraints was also tested. When changes are applied to the same table, correct succession is assured by the order of logs in each databases' query log file, which is read by Debezium and because Kafka consumers always read messages from a topic in order. The mechanism described in Application description section guarantees that entities are inserted into MySQL in the correct sequence and that no data are lost. If a row that has missing dependencies is inserted, this throws an error which is handled by waiting for an amount of time for the dependencies to be inserted by the other threads and then the insertion is retried. For each test, the databases were reset, i.e., all data were deleted. For modification and deletion, the tests started from the same state. Data for the tests were generated using the *com.github.java.faker* library.

The test runs were triggered by performing a GET on the tests' corresponding endpoint. After performing tests for MongoDB, the databases were purged.

All the tests presented were conducted on a computer with the following configuration: Windows 10 Pro 64-bit, Intel Core processor i9-9900K CPU@3.60 GHz, 16 GB RAM, and a 512 GB SSD being used for MySQL version 8.0.32 and for MongoDB version 6.0.4. The networking hardware is not specified since all the systems were deployed on the same machine.

### 4.1. Latency Testing

Data latency was measured by determining the time that elapsed from the time an insert was made in one database to the time that the change arrived in the other database. Tests were performed on both sides.

For a change to be carried out from MongoDB to MySQL, the average time in milliseconds obtained by repeating, is 89 ms; the other way, from MySQL to MongoDB, it is 93.5 ms. The time obtained as a result of the individual tests performed is shown in Figure 4. For the times obtained, which are under 100 milliseconds, it can be said that no latencies that can be felt by a user were introduced for the vast majority of existing applications.

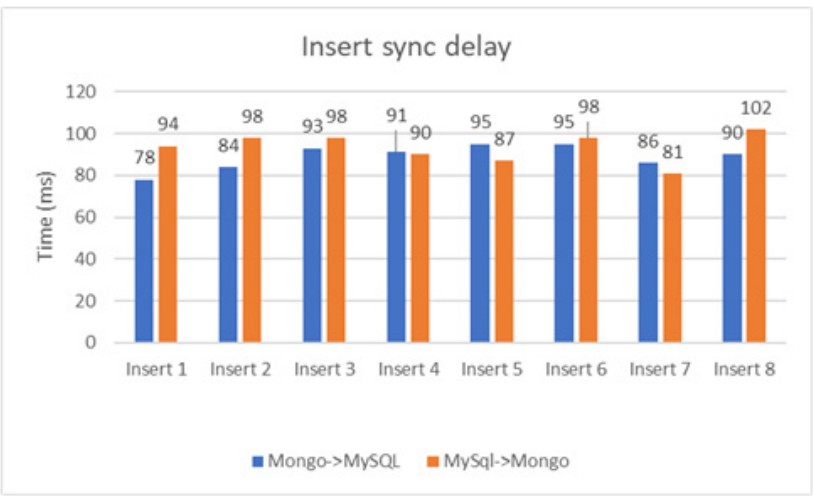

**Figure 4.** Insert synchronization delay for both directions.

*4.2. Data Consistency*

In the process of synchronizing the two databases, avoiding constraint errors, especially the foreign key ones, in the relational database represents a major issue.

The proper functioning of this mechanism was tested by inserting rows in the *passenger* table, rows in the *flight* table, and rows in the *ticket* table. The ticket table contains foreign keys that refer to the primary keys to the *passenger* and *flight* tables. Starting with empty databases, the data were inserted initially into MongoDB, then the existence of the data in MySQL was checked by checking the foreign key constraints, and the delay was recorded.

For each table, the primary keys were assigned sequentially to the to-be-inserted entities such that no primary key constraint violations would arise during testing: from 1 to n, where *n* is the number of rows to be inserted for each test. The tables were purged between tests.

This test demonstrates the ability to keep the data consistent by avoiding foreign key constraint violations. Moreover, this test also measures the performance when inserting multiple rows.

First, it was observed that all the rows inserted in MongoDB were also inserted into MySQL, there were no missing data. Some rows from the tickets table were tentatively inserted before the rows it depended on. The retry mechanism handled this successfully. Analyzing the data in Figure 5, it can be seen that for under 100 rows for each table, the sync time is similar and below one second. For a number of rows greater than 100, the delays seem to grow linearly.

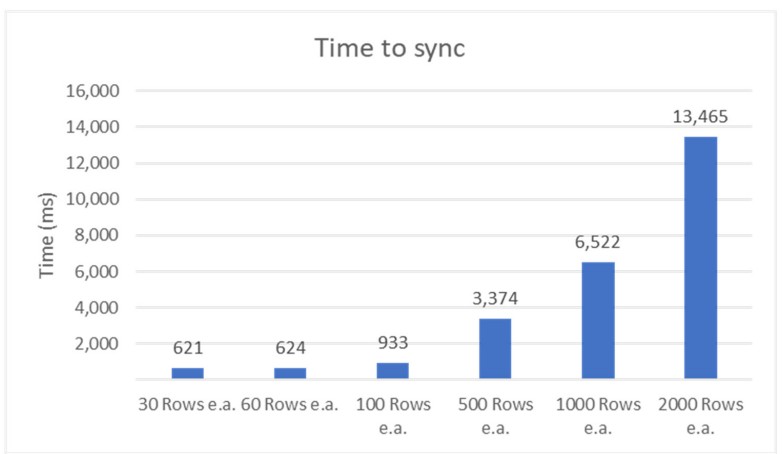

**Figure 5.** Time until the two databases are in sync after inserting multiple rows in different tables bound by FK constraints.

### 4.3. Operation Replication Performance

Performance was tested for the following operations: insert, update and delete. The tests were two-way, i.e., the performance of replicating changes from MongoDB to MySQL was measured as well as the performance of replicating changes from MySQL to MongoDB. All tests were performed with 6 data sets. The essential difference between the data sets was given by the number of records. The tests were carried out with 30, 60, 100, 500, 1000 and 2000 records, respectively. In the code, these values are found in an array called *testSizes*. The testing procedure is detailed in Algorithms 1–3 for each of the operations tested. The tests follow the same pattern: the data are prepared, the operation is executed in the first database, and a timer is started. When the synchronization is complete, as noted by a continuous polling of the second database, the timer is stopped and the result is printed. Basically, the time-to-sync that is recorded is the time between the end of the execution of an operation in the first database and the moment its effect is mirrored in the second database. The polling interval is set to 20 ms using the static variable, POLLING_SLEEP_TIME_IN_MS. The tests were run multiple times to ensure that the results were consistent.

#### 4.3.1. Insert Operation

The insertion operation for MySQL and MongoDB was performed as shown in Algorithm 1.

---

**Algorithm 1.** Insert operations.

**MongoDB Insert Operation**

```
@GetMapping("/testMongoBulkInsert")
  String testMongoBulkInsert() throws InterruptedException {
    // First, we prepare the data to be inserted into MongoDB
    List<Map<String, Object>> objects = new ArrayList<>();
    int id = 0;
    for (int i = 0; i < testSizes.length; i++) {
        objects = new ArrayList<>();
        for (int j = 0; j < testSizes[i]; j++) {
            id++;
            Map<String, Object> entity = new HashMap<>();
            entity.put("_id", id);
            entity.put("name", faker.name().firstName());
            entity.put("surname", faker.name().lastName());
            entity.put("SSN", faker.number().digits(9));
            entity.put("age", faker.number().numberBetween(1,80));

            objects.add(entity);
        }
        // Here, we insert the data using bulk operations
        BulkOperations bulkOps = mongoTemplate.bulkOps(BulkMode.UNORDERED,
"passenger");
        for (Map<String, Object> document: objects) {
            bulkOps.insert(document);
        }
        bulkOps.execute();
        // Here, we poll MySQL to detect the synchronization timing
        long start = System.currentTimeMillis();
        String query = "select count(*) from" + "passenger";
        Integer res = 0;
        while (res != finalCount[i]) {
            Thread.sleep(POLLING_SLEEP_TIME_IN_MS);
```

**Algorithm 1.** *Cont.*

```
            res = jdbcTemplate.queryForObject(query, Integer.class);
            if (res == null) {
                res = 0;
            }
        }
        System.out.println("done" + testSizes[i] + "in" + (System.currentTimeMillis() − start));
    }
    return "done";
}
```

**MySQL Insert Operation**

```
@GetMapping("/testSqlBulkInsert")
  String testSqlBulkInsert() throws InterruptedException {
      // First, we prepare the data to be inserted into MySQL
      List<Object[]> objects = new ArrayList<>();
      int id = 0;
      for (int i = 0; i < testSizes.length; i++) {
          objects = new ArrayList<>();
          for (int j = 0; j < testSizes [i]; j++) {
              id++;
              Object[] o = new Object[5];
              o[0] = id;
              o[1] = faker.name().firstName();
              o[2] = faker.name().lastName();
              o[3] = faker.number().digits(9);
              o[4] = faker.number().numberBetween(1, 80);
              objects.add(o);
          }
          // Here, we insert the data into MySQL using a batch statement
          String sql = "INSERT INTO passenger (id, name, surname, ssn, age) VALUES (?, ?, ?, ?,
?)";
          jdbcTemplate.batchUpdate(sql, objects);
          // Here, we poll MongoDB to detect the synchronization timing
          long start = System.currentTimeMillis();
          Query q = new Query();
          long count = mongoTemplate.count(q, "passenger");
          while (count != finalCount[i]) {
              Thread.sleep(POLLING_SLEEP_TIME_IN_MS);
              count = mongoTemplate.count(q, "passenger");
          }
          System.out.println("done" + testSizes[i] + "in" + (System.currentTimeMillis() − start));
      }
      return "done";
  }
```

Analyzing the results for the insert operation shown in Figure 6, it can be noted that the time needed to synchronize the two databases is similar in both parts; even if the discrepancy increases with the number of entries, it is not an impediment to the proper operation of an application that uses the synchronization system described in this paper. For a small number of entries, i.e., 30 or 60, the times are similar. This shows us that there is a minimum required processing time; even if the latency for one input is under 100 milliseconds, from a certain number of inserted records, the processing time grows linearly.

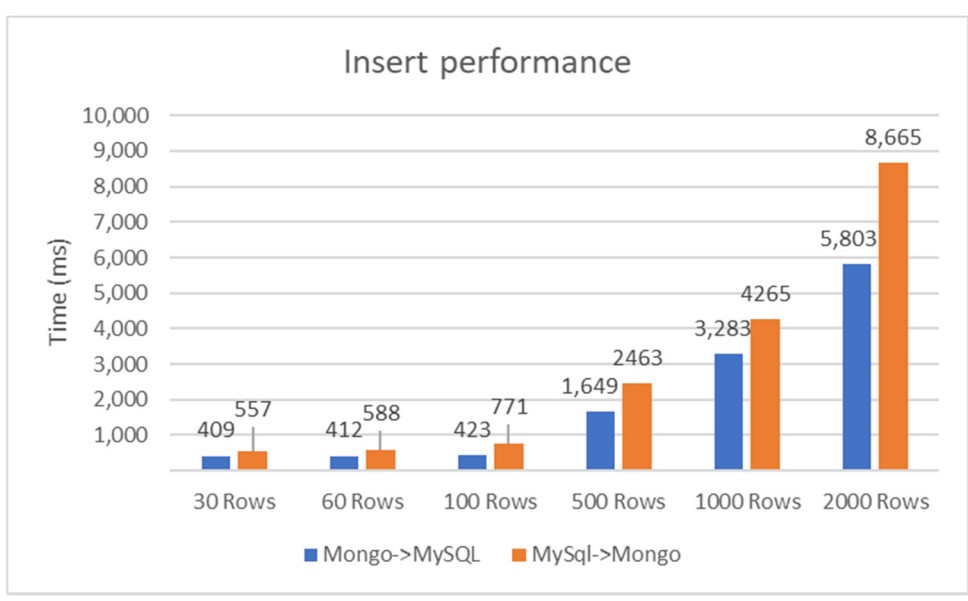

**Figure 6.** Execution times for the insert operation showing the source and destination of the change.

The ratio between the time to synchronization and the number of rows/entries, as can be seen in Figure 7, decreases depending on the number of entries. For a few entries, we have an unfavorable ratio and then the performance increases and the ratio reach a value of 2.9 ms/entry for MongoDB to MySQL replication, and for MySQL to MongoDB replication, the ratio reaches a value of 4.3 ms/entry.

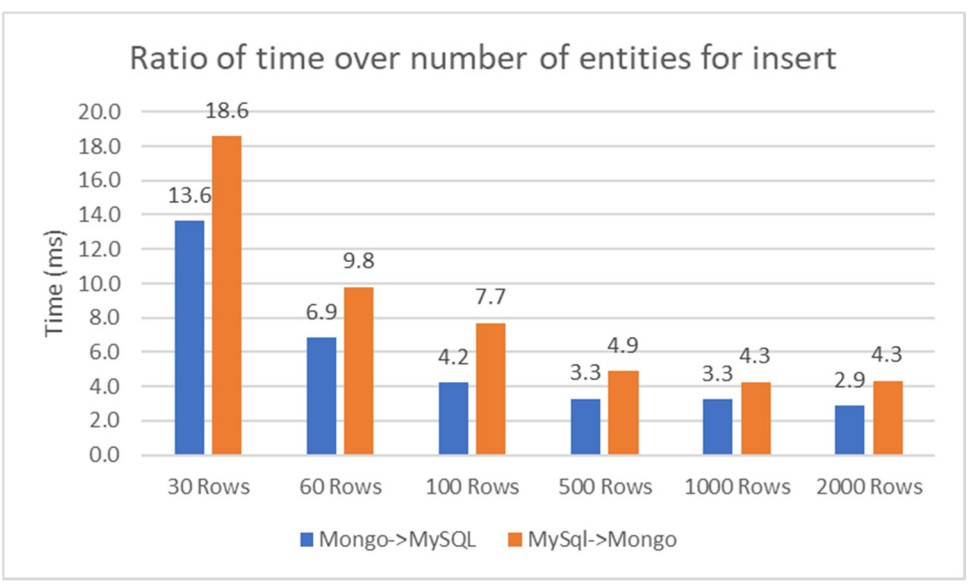

**Figure 7.** Ratio of time over number of rows for insert.

### 4.3.2. Update Operation

The update operation for MySQL and MongoDB was performed as shown in Algorithm 2.

---

**Algorithm 2.** Update operations.

---

**MongoDB Update Operation**

---

```
@GetMapping("/testMongoUpdate")
   String testMongoUpdate() throws InterruptedException {
      // First, we prepare the data to be updated into MongoDB
      int id = 0;
      for (int i = 0; i < testSizes.length; i++) {
         BulkOperations bulkOps = mongoTemplate.bulkOps(BulkMode.UNORDERED,
"passenger");
         for (int j = 0; j < testSizes[i]; j++) {
            id++;
            Query query = new Query();
            query.addCriteria(Criteria.where("_id").is(id));
            Update update = new Update();
            update.set("name", "Anthony");
            bulkOps.updateOne(query, update);
         }
         // Here, we execute the update in MongoDB using a batch statement
         bulkOps.execute();
         // Here, we poll MongoDB to detect the synchronization timing
         long start = System.currentTimeMillis();
         String query = "select count(*) from" + "passenger where name = "Anthony";
         Integer res = 0;
         while (res != finalCount[i]) {
            Thread.sleep( POLLING_SLEEP_TIME_IN_MS);
            res = jdbcTemplate.queryForObject(query, Integer.class);
            if (res == null) {
               res = 0;
            }
         }
         System.out.println("done" + testSizes[i] + "in" + (System.currentTimeMillis() − start));
      }
      return "done";
   }
```

---

**MySQL Update Operation**

---

```
@GetMapping("/testSqlUpdate")
  String testSqlUpdate() throws InterruptedException {
     // First, we prepare the data to be updated in MySQL
     for (int j = 0; j < testSizes.length; j++) {
        int[] wj = new int[]{j}; // only final(non-modifiable) variables allowed in lambdas, we
used a single value array to bypass the restriction
        String sql = "UPDATE passenger SET name = ? WHERE id = ?";
        // Here, we update the data in MySQL using a batch statement
        jdbcTemplate.batchUpdate(sql, new BatchPreparedStatementSetter() {
           @Override
           public void setValues(PreparedStatement preparedStatement, int i) throws
SQLException {
              int myId;
              if (wj[0] == 0) {
                 myId = i + 1;
              } else {
                 myId = finalCount[wj[0] - 1] + i + 1;
              }
              preparedStatement.setString(1, "Kelly");
              preparedStatement.setInt(2, myId);
           }

           @Override
```

---

---

**Algorithm 2.** *Cont.*

---

```
        public int getBatchSize() {
            return testSizes[wj[0]];
        }
    });
    // Here, we poll MongoDB to detect the synchronization timing
    long start = System.currentTimeMillis();
    Query q = new Query(Criteria.where("name").is("Kelly"));

    long count = mongoTemplate.count(q, "passenger");
    while (count != finalCount[j]) {
        Thread.sleep( POLLING_SLEEP_TIME_IN_MS);
        count = mongoTemplate.count(q, "passenger");
    }
    System.out.println("done" + testSizes[j] + "in" + (System.currentTimeMillis() − start));
  }
  return "done";
}
```

---

When using MongoDB, there are different methods for modifying one or more items, as well as inserting them. Multiple entries should be updated in batch. We constructed a query that does this. First, we add a criterion to the query, functioning as a filter, through which the elements to be modified are obtained, then we set the name of the field to be updated and provide the update value. Another approach can be seen in [25].

Analyzing the results shown in Figure 8, it can be observed that these are very similar to those of the insert operation. The ratio between the time to synchronization and the number of entries also follow the pattern seen in measuring the insert synchronization performance (Figure 9).

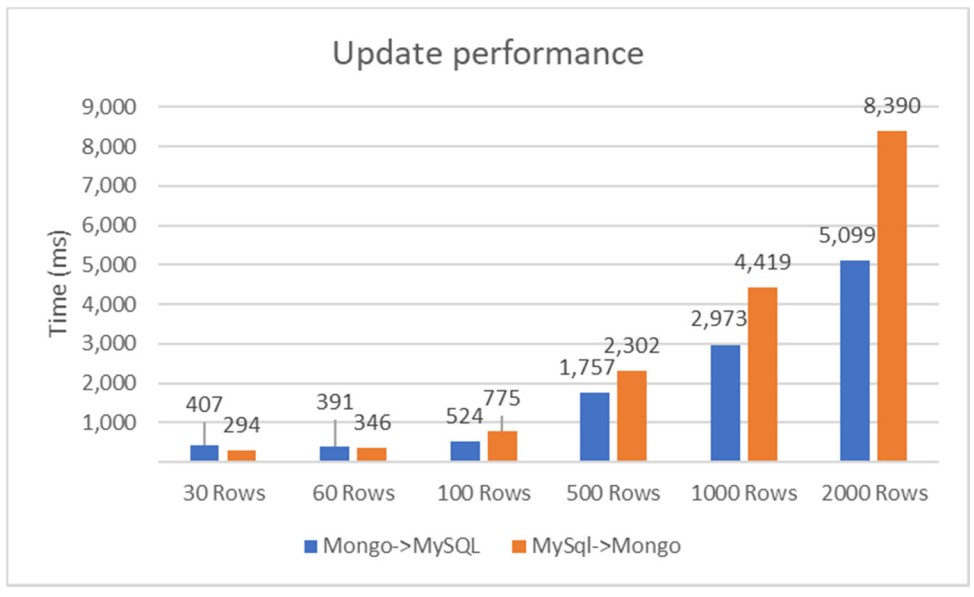

**Figure 8.** Execution times for the update operation showing the source and destination of the change.

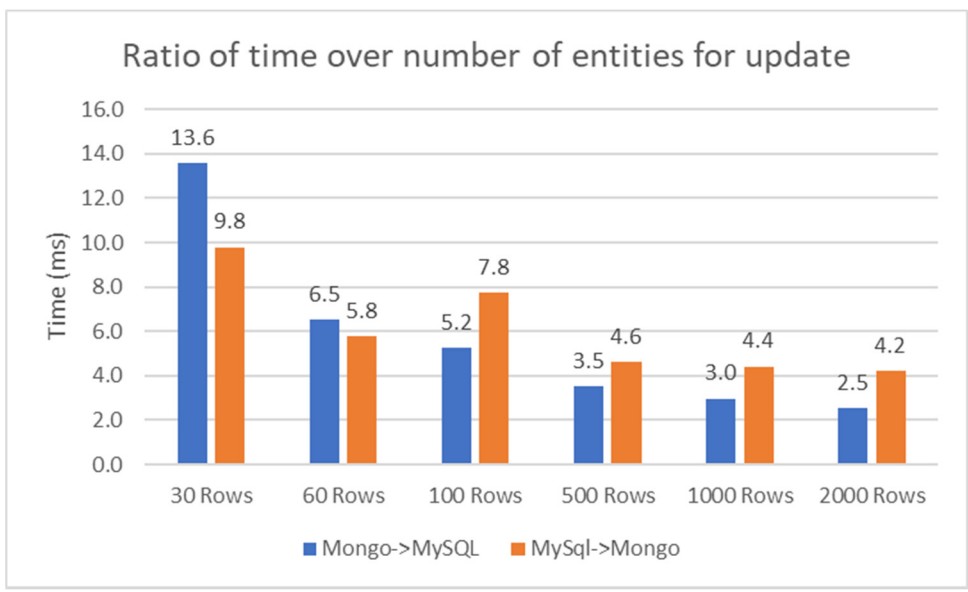

**Figure 9.** Ratio of time over number of rows for update.

### 4.3.3. Delete Operation

The delete operation for MySQL and MongoDB was performed as shown in Algorithm 3.

---

**Algorithm 3.** Delete operation.

**MongoDB Delete Operation**

```
@GetMapping("/testMongoDelete")
  String testMongoDelete() throws InterruptedException {
    // First, we prepare the data to be deleted from MongoDB
    int id = 0;
    List<Integer> ids;
    for (int i = 0; i < testSizes.length; i++) {
        ids = new ArrayList<>();
        for (int j = 1; j <= testSizes[i]; j++) {
            id++;
            ids.add(id);
        }
        Query query = Query.query(Criteria.where("_id").in(ids));
        // Here, we delete the data from MongoDB using a single query
        mongoTemplate.remove(query, "passenger");

        long start = System.currentTimeMillis();
        // Here, we poll MySQL to detect the synchronization timing
        String queryString = "select count(*) from" + "passenger";
        Integer res = −1;
        while (res != finalCountForDelete[i]) {
            Thread.sleep(POLLING_SLEEP_TIME_IN_MS);
            res = jdbcTemplate.queryForObject(queryString, Integer.class);
            if (res == null) {
                res = 0;
            }
        }
        System.out.println("done" + testSizes[i] + "in" + (System.currentTimeMillis() − start));
    }
    return "done";
  }
```

| **Algorithm 3.** *Cont.* |
| --- |
| **MySQL Delete Operation** |

```
@GetMapping("/testSqlDelete")
    String testSqlDelete() throws InterruptedException {
        // First, we prepare the data to be deleted from MySQL
        for (int j = 0; j < testSizes.length; j++) {
            String sql = "DELETE FROM passenger WHERE id = ?";
            int[] wj = new int[]{j}; // only final(non-modifiable) variables allowed in lambdas, we
used a single value array to bypass the restriction
            // Here, we delete the data from MySQL using a batch statement
            jdbcTemplate.batchUpdate(sql, new BatchPreparedStatementSetter() {
                @Override
                public void setValues(PreparedStatement preparedStatement, int i) throws
SQLException {
                    int myId;
                    if (wj[0] == 0) {
                        myId = i + 1;
                    } else {
                        myId = finalCount[wj[0] − 1] + i + 1;
                    }
                    preparedStatement.setInt(1, myId);
                }
                @Override
                public int getBatchSize() {
                    return testSizes[wj[0]];
                }
            });
            // Here, we poll MongoDB to detect the synchronization timing
            long start = System.currentTimeMillis();
            Query q = new Query();

            long count = mongoTemplate.count(q, "passenger");
            while (count != finalCountForDelete[j]) {
                Thread.sleep(POLLING_SLEEP_TIME_IN_MS);
                count = mongoTemplate.count(q, "passenger");
            }
            System.out.println("done" + testSizes[j] + "in" + (System.currentTimeMillis() − start));
        }
        return "done";
    }
```

The deletion of a passenger is performed using predefined methods—command *delete* when using MySQL, and *remove()* method when using a MongoDB database, as shown in Algorithm 3.

Analyzing the results shown in Figure 10 for the deletion operation, it can be seen that the synchronization times obtained in the case of the deletion operation are similar in both directions for MySQL and MongoDB. Thus, we can say that a favorable result was obtained. This ensures that the synchronization system described in this paper can be used in a real-world application. For a small number of entries, i.e., 30 or 60, the times are similar. This shows that there is a minimum required processing and deletion time. The ratio between the time to synchronization and the number of entries decreases according to the number of entries, as shown in Figure 11. For a few entries, we have an unfavorable ratio but afterwards, the performance increases and tends towards 2.7 ms/entry for MongoDB to MySQL replication and to 4.3 ms/entry for MySQL/MongoDB.

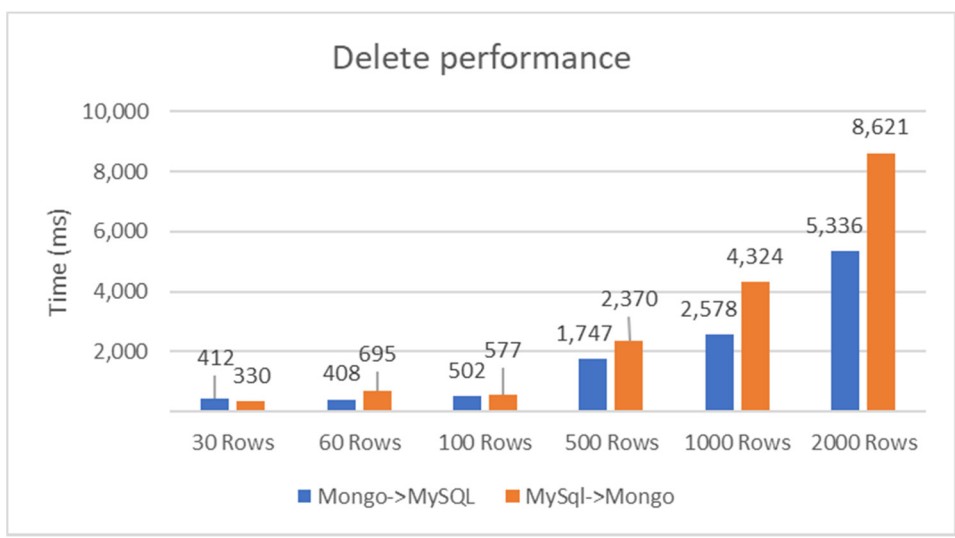

**Figure 10.** Execution times for the delete operation showing the source and destination of the change.

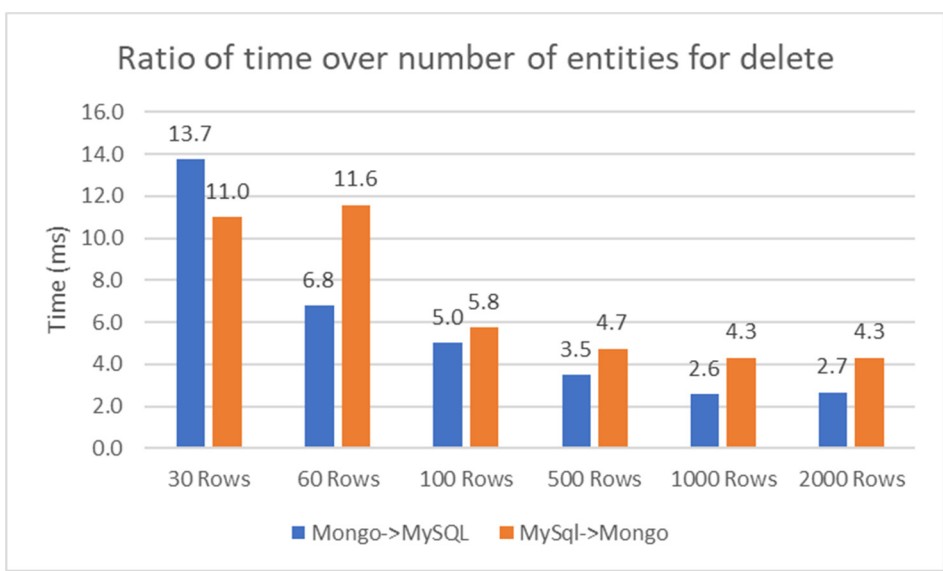

**Figure 11.** Ratio of time over number of rows for delete.

### 5. Conclusions

In this paper, a synchronization method between relational and non-relational databases was presented. The proposed approach was tested by creating an application in Java that is based on Spring Boot that uses Spring Data. The practical tests that were carried out have shown how this synchronization can be achieved in the case of tables with relationships between them for different CRUD operations.

The paper has explored the method and technology behind the proposed synchronization method between MySQL as a relational database and MongoDB as a non-relational database, showing that that synchronization between the two databases, MySQL and MongoDB, is not only possible but also efficient.

As revealed in the study, the tests carried out have highlighted that the synchronization between MySQL as a relational database and MongoDB as a non-relational database occurs with an acceptable delay.

The present work is a proof of concept that can be the basis of a complete synchronization solution, which could also take into account the more complex functionalities of a relational database, such as triggers.

The solution proposed in this paper, due to the fact that the Debezium CDC (Change Data Capture) system was used, can be easily extended to provide synchronization between other databases such as PostgreSQL, SQL Server, Db2, Cassandra, Vitess, and Spanner.

Although a solid foundation has been laid for achieving synchronization between a relational and a non-relational database, there are several improvements that can be made to the system to make it faster, more robust, or more applicable. For example, in the situation where more than 1000 entities are created and immediately deleted, an infinite processing cycle arises because the checks implemented in code do not foresee this case. A solution would be to insert the entities' IDs into a collection where entries that have been added and deleted in the last 5 min are kept and, before executing an insert, check if the to-be-inserted ID exists in that collection.

In the application developed, when a new table is created in MySQL, it is not created automatically in MongoDB, so its topic will not be listened to. The list of topics should be periodically checked and consumers should be instantiated for new topics. In the case of deleting a table or collection, it is necessary to query the database directly and, if it is found that the table is missing, stop the consumer. If new fields are added to MongoDB, the application will throw an exception. In this case, its data type should be determined and the structure of the MySQL table altered to include a column matching the field's type.

Some SQL tables may not have an id field. The developed application does not handle this case. In future iterations, to handle this case, the primary key field needs to be determined and an index created for that field in MongoDB. In the case of a composite primary key, a solution is to create a composite index for those columns. For UNIQUE fields with other names than an ID, since Debezium does not provide information about constraints, the application will have to check for them when a table is created or when an ALTER TABLE statement is executed and triggers a message on the schema topic. After it detects the unique column, it can create an index in MongoDB for that column.

The tests showed good performance and reliability in the synchronization operations. The application is capable of processing moderate volumes of data, which means that it could be used in many real-world scenarios where synchronization is needed, though its applicability may depend on the specifics of each project. It should also be noted that setting up synchronization between MySQL and MongoDB requires a solid understanding of each database's data structures.

Consequently, the proposed synchronization method between MySQL and a MongoDB database is a viable and efficient process. The proposed solution is not only capable of synchronizing a relational database with a non-relational one but is performant and feasible to use in real scenarios.

**Author Contributions:** Conceptualization, C.A.G., T.T. and D.R.Z.; methodology, C.A.G., T.T. and D.R.Z.; software, T.T. and R.Ș.G.; validation, C.A.G. and R.Ș.G.; writing—original draft preparation T.T., D.R.Z. and C.A.G.; writing—review and editing, C.A.G. and R.Ș.G.; All authors have read and agreed to the published version of the manuscript.

**Funding:** The publication of this research was supported by the University of Oradea.

**Data Availability Statement:** Not applicable.

**Conflicts of Interest:** The authors declare no conflict of interest.

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
