# Peer review of "Implementing a Synchronization Method between a Relational and a Non-Relational Database"

_2504-2289, doi:10.3390/bdcc7030153_

Round 1

Reviewer 1 Report

The paper is based upon adequately address the management of synchronization between a relational and a non-relational (NoSQL) database. The paper needs to discuss the previous proposed literature conducted in correspondence to the novelty of the work done so far. 

Also, the paper discuss about the algorithm designed and implemented also in the past several synchronization are implemented how different is the new algorithm with the previous studies

paper needs a proof reading throughout grammatical  errors are existing 

The paper is based upon adequately address the management of synchronization between a relational and a non-relational (NoSQL) database. The paper needs to discuss the previous proposed literature conducted in correspondence to the novelty of the work done so far. 

Also, the paper discuss about the algorithm designed and implemented also in the past several synchronization are implemented how different is the new algorithm with the previous studies

paper needs a proof reading throughout grammatical  errors are existin

Reviewer 2 Report

The topic is interesting.

In the abstract the authors say "This paper proposes a solution to this problem". An adjective to the solution should be added. IT solution, conceptual solution, or you will decide the appropriate work.

The authors have to be precise "synchronization between a relational and a non-relational database". So the authors try to sync databases and/or DBMS? This issue should be clear in the whole text. Sometimes the authors speak about "databases" but they mean DBMS. Please, read the whole article carefully and check.

"Because relational databases could not handle large volumes of data and process them instantly" It is not true. I see there is a citations. Please, revise. Add "According to some authors...".

The authors use many technologies (JAVA, Kafka, Spring boot,...). Could the solution be reached with less technologies and less efforts?

Fig. 3 gives MongoDB database structure. Add "date of the flight", "flight number".

Do the authors offer automatic synchronization?

was -> is

were -> are

Use Passive voice where possible. E.g., We used Debezium -> Debezium is used.

The performance testing is useful.

It is good that programming code is given.

Reviewer 3 Report

- My first concern is the lack of motivation for the need of this synchronization between noSQL and relational databases. The authors start with presenting their solution without explaining what is exactly is the problem.
- According to ithenticate system (plagiarism detection system) 6% of this work is extracted from another document published by the same authors (Performance Impact of Optimization Methods on MySQL Document-Based and Relational Databases). A some extracts are inserted without any changes.
-I would like to see the executable code described in this paper.
It would be practical if other scientists could also try this tool and if the code were available on some code repository then they could contribute their ideas. If that code cannot be open source, then it would be nice to have a website where you can try the tool.
- There is a lack of relevant references especially in the introduction.
- I am not an expert in English, but it seems to me that some sentences are not well composed.

Round 2

Reviewer 3 Report

Thank you. I suggest to accept this paper in current form